# Structure-based evolution of a promiscuous inhibitor to a selective stabilizer of protein–protein interactions

Eline Sijbesma [1,5], Emira Visser[1,5], Kathrin Plitzko[2], Philipp Thiel[3], Lech-Gustav Milroy[1], Markus Kaiser [2✉], Luc Brunsveld [1✉] & Christian Ottmann[1,4✉]

The systematic stabilization of protein–protein interactions (PPI) has great potential as innovative drug discovery strategy to target novel and hard-to-drug protein classes. The current lack of chemical starting points and focused screening opportunities limits the identification of small molecule stabilizers that engage two proteins simultaneously. Starting from our previously described virtual screening strategy to identify inhibitors of 14-3-3 proteins, we report a conceptual molecular docking approach providing concrete entries for discovery and rational optimization of stabilizers for the interaction of 14-3-3 with the carbohydrate-response element-binding protein (ChREBP). X-ray crystallography reveals a distinct difference in the binding modes between weak and general inhibitors of 14-3-3 complexes and a specific, potent stabilizer of the 14-3-3/ChREBP complex. Structure-guided stabilizer optimization results in selective, up to 26-fold enhancement of the 14-3-3/ChREBP interaction. This study demonstrates the potential of rational design approaches for the development of selective PPI stabilizers starting from weak, promiscuous PPI inhibitors.

[1] Laboratory of Chemical Biology and Institute for Complex Molecular Systems (ICMS), Department of Biomedical Engineering, Eindhoven University of Technology, Eindhoven, Netherlands. [2] Chemical Biology, Center of Medical Biotechnology, Faculty of Biology, University of Duisburg-Essen, Duisburg, Germany. [3] Institute for Biomedical Informatics and Medical Informatics, University of Tübingen, Tübingen, Germany. [4] Department of Organic Chemistry, University of Duisburg-Essen, Duisburg, Germany. [5] These authors contributed equally: Eline Sijbesma, Emira Visser. ✉email: markus.kaiser@uni-due.de; l.brunsveld@tue.nl; c.ottmann@tue.nl

Proteins interact with other proteins to exert their physiological functions in the context of complex spatiotemporally distributed protein–protein interaction (PPI) networks[1,2]. PPIs are attractive drug targets due to their essential regulation of nearly all cellular processes and, as such, PPI modulation has a vast therapeutic potential[3–6]. In fact, the inhibition of PPIs has rapidly evolved to the frontlines of modern drug discovery and has significantly extended the druggable genome[7,8]. However, the opposite strategy of PPI enhancement by small molecule stabilizers is underexplored, when in fact this strategy offers unique advantages due to the uncompetitive nature of stabilizers and specificity for a transient complex over the individual proteins[9–12].

Whereas immunosuppressants rapamycin, cyclosporine, and FK506, and the antitumor drug paclitaxel have been long used in the clinic[8–10], interest in PPI stabilization as a conceptual strategy has only recently surged, due to the success of synthetically engineered hetero-bifunctional probes (proteolysis-targeting chimera; PROTACs)[13,14] and the revelation of the molecular mechanism of lenalidomide and thalidomide (immunomodulatory drugs; IMiDs®) as PPI stabilizers[15,16]. Nevertheless, the majority of reported PPI stabilizers have been serendipitous discoveries and systematic design, screening, and technology platforms for PPI stabilizer discovery are largely lacking[17,18]. There is thus an urgent need for conceptual strategies for hit finding and rational optimization, empowering PPI stabilization.

The PPI of 14-3-3 with the carbohydrate-response element-binding protein (ChREBP) regulates transcription of glucose-responsive genes. Whereas most of the 14-3-3 clients require to be phosphorylated prior to 14-3-3 binding[19,20], ChREBP is one of the very few phosphorylation-independent 14-3-3 partner proteins[21–23] and interacts with 14-3-3 in a unique α-helical conformation (residues 117–137)[24]. A free sulfate or phosphate in the 14-3-3 phospho-accepting pocket interacts with both proteins[24]. In addition, adenosine monophosphate (AMP) has been reported to bind this pocket, thereby mildly stabilizing the PPI complex and enhancing 14-3-3's regulation of ChREBP cytosol-nuclear trafficking[25]. Novel ChREBP/14-3-3 stabilizers could be valuable regulators of this glucose-responsive transcription factor.

Structure-based in silico approaches have proven their value in classical drug discovery[26–30]. Here, a structure-guided virtual screening and molecular docking cascade, employing a known PPI inhibitor class as starting point, is brought forward as a strategy for the identification of PPI stabilizers[3]. We report on a successful in silico screening strategy for stabilization of native PPIs via ligands with a molecular glue mode of action. The starting point for this approach is a screening methodology for the identification of inhibitory phosphonates/phosphates that bind to the phosphoserine/-threonine binding pocket in 14-3-3 and block 14-3-3 PPIs in a widespread manner[31]. Small-molecule stabilizers of the ChREBP/14-3-3 protein complex are identified that indeed engage a composite interface pocket constituted by both protein partners. Our structure-based optimization and two high-resolution X-ray crystal structures reveal a distinct difference in binding modes, enabling stabilatory and weak inhibitory activity of a common phosphonate scaffold to be entirely disconnected, resulting in up to 26-fold and selective PPI stabilization without significant PPI inhibition. These findings thus illustrate the power of this rational approach for future PPI stabilization-based drug discovery.

## Results

### Docking reveals small-molecule 14-3-3/ChREBP stabilizers.
The importance of the phospho-group—both in 14-3-3-binding motifs and PPI inhibitors—directed the selection of chemical starting points to molecules that bind the phospho-accepting pocket of 14-3-3. Phosphate- and phosphonate-based inhibitors typically inhibit 14-3-3/client complexes in the low micro-molar ($IC_{50} \sim 1–20$ μM) range[31–33]. It has previously been shown that physiological levels of phosphate anions can furthermore affect the 14-3-3 phospho-interactome, via concentration-dependent dissociation of 14-3-3/client complexes[34]. Whereas phosphate- and phosphonate-containing moieties thus generally compete with 14-3-3 target binding, for the 14-3-3/ChREBP complex the phospho-binding pocket is uniquely positioned at the rim of the interface, presenting an opportunity for phosphate-/phosphonate-based PPI stabilization. The two crystal structures of 14-3-3/ChREBP in the Protein Data Bank (PDB; entries 4GNT and 5F74) served as entry points for the structure-based in silico screen. A phospho-binding pocket-centered receptor grid was generated for the structure of 14-3-3β bound to the α2 helix of ChREBP (Fig. 1a). The first step of the virtual screening procedure selected for a phosphate or phosphonate group by a substructure filter which yielded 869 virtual compounds (of the initial 5,993,085 in the public MolPort database) (Fig. 1b). After additional selection filters for drug-like properties, 471 compounds were subjected to molecular docking into the receptor grid using Glide[35,36]. Hits were additionally docked into the receptor using an induced fit docking protocol, taking conformational changes of amino acid side chains in the active site into account[37,38]. We selected 13 compounds for in vitro testing from the 200 top-ranked docking poses, based on visual inspection, divided among three distinct subclasses; AMP-like structures (class A); and non-AMP-like phosphates (B) and phosphonates (C; Supplementary Tables 1–3 and Supplementary Figs. 1–3). Class A including AMP itself did not show stabilization of the 14-3-3/ChREBP interaction in a fluorescence anisotropy assay (Supplementary Fig. 4). We did observe stabilization by AMP based on ITC data (Supplementary Fig. 5), validating it as a positive control and in line with literature. In both B- and C subclasses one hit was found to increase 14-3-3/ChREBP binding (**1** and **2** with $EC_{50}$ values of 0.7 and 45 μM, and ligand efficiency (LE) of 0.28 and 0.32, respectively; Supplementary Fig. 4). The docking poses for **1** and **2** revealed that their phosphate or phosphonate groups were indeed ideally positioned in the basic cavity, constituted by the 14-3-3 Arg–Arg–Tyr phospho-accepting triad, and interacting with the tryptophan side chain of ChREBP (Fig. 1c). **1** and **2** increased the binding affinity of 14-3-3β for ChREBP in a dose-dependent fashion up to 10- and 4-fold, respectively (Fig. 1d).

### Nearest neighbor analysis yields an improved stabilizer.
The more attractive and synthetically more accessible phosphonate-based scaffold of **2**, as compared with the reactive phosphate **1**, prompted its chemical optimization to establish a structure activity relationship (SAR). An initial SAR-by-catalog study of eight compounds resulted in an increased PPI stabilization by **3** ($EC_{50}$ 5.2 μM; Fig. 2a, b and Supplementary Table 4), with a 14-fold enhancement of the binding affinity of 14-3-3β for ChREBP (Fig. 2c). Shorter linkers to the second phenyl group, without an amide, were inactive (**4–8**), as was phenylphosphate (**9**) and a weak inhibitory effect was observed for phenylphosphonate (**10**) (Fig. 2b). Interestingly, phenylphosphonate-based scaffolds had also surfaced as hits for 14-3-3 in a virtual screen reported by us previously[31]. Whereas the focus of that work was on finding disruptors of the interaction between 14-3-3 and aminopeptidase N (APN), the target pocket appears identical. The fundamental differences between a PPI disruptor—that needs to tightly bind its target protein to compete with protein complex formation, and a PPI stabilizer—that binds a specific pocket at a PPI interface, lie at the basis of potential selectivity for stabilization, especially

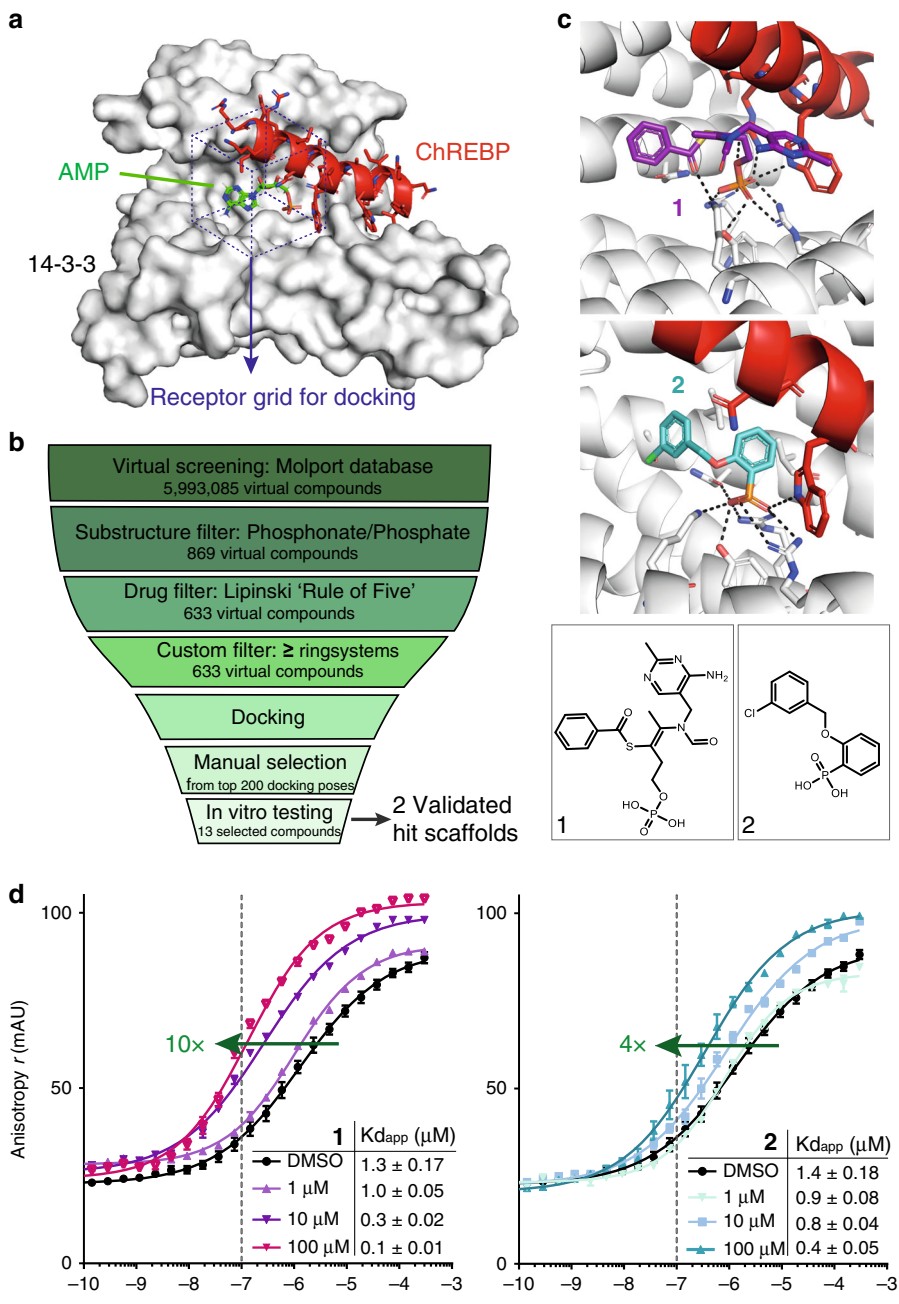

**Fig. 1 A structure-based in silico screen for small-molecule stabilizers of the 14-3-3/ChREBP protein complex. a** The receptor grid (purple dotted box) for docking in the crystal structure of 14-3-3β (gray surface), ChREBP (red cartoon and sticks), and AMP (green sticks) (PDB entry 5F74)[25]. **b** Overview of the virtual screening procedure. **c** Docking poses and chemical structures of **1** and **2**. **d** Titration curves of 14-3-3β on 100 nM fluorescently-labeled ChREBP peptide in the presence of increasing concentrations (1, 10, 100 μM) of **1** or **2**. Data and error bars represent mean ± SEM, n = 3 replicates. Source data are provided as a Source data file.

when targeting promiscuous PPI pockets. We hypothesized this also to be the case for the protein complex constituted of 14-3-3 and ChREBP and aimed to explore selective stabilization of this PPI by exploiting the phospho-pocket at its composite interface.

**Crystal structure elucidates molecular glue mode of action.** We thus set out to study the molecular mechanism and optimize the stabilizing activity of **3** by obtaining structural insights of its mode of action. The tertiary co-crystal structure of 14-3-3β bound to **3** and the ChREBP peptide was solved by X-ray

crystallography (Fig. 3 and Table 1). The overall complex resembles the previously reported crystal structures for a 14-3-3β dimer with the two antiparallel-binding ChREBP-α2 helices. **3** was indeed found clearly positioned in the phospho-accepting pocket of 14-3-3, interacting with R128 of ChREBP, and K51, R58, R129, and Y130 of 14-3-3 (Fig. 3c). A relevant, additional intramolecular polar interaction was observed for **3** between its amide nitrogen and a phosphonate oxygen, stabilizing its protein-bound state geometry. The phenyl of **3** on one side faces an ensemble of hydrophobic residues of both ChREBP (I120) and 14-3-3 (L218, I219, L174, and L222). R128 of ChREBP 'bridges'

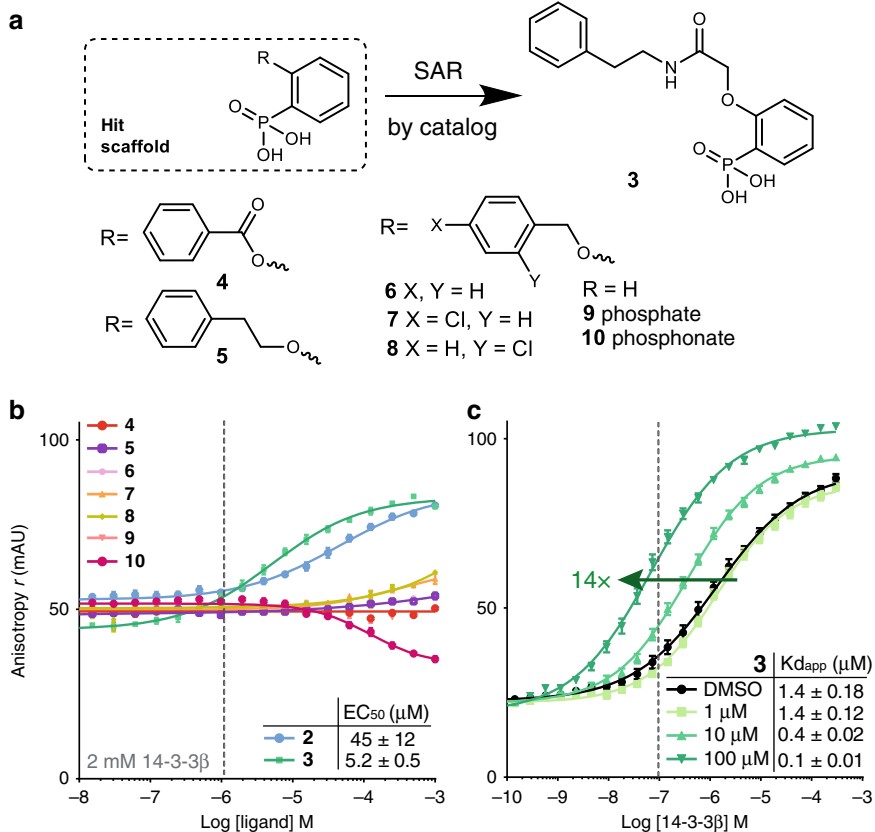

**Fig. 2 Initial optimization via SAR-by-catalog of the hit scaffold 2-based series. a** Eight derivatives of **2** were selected for molecular docking and in vitro testing. Titration curves of **b** ligands on 2 μM 14-3-3β and 100 nM fluorescently-labeled ChREBP peptide, and **c** of 14-3-3β on 100 nM fluorescently-labeled ChREBP peptide in presence of increasing concentrations (1, 10, 100 μM) of compound **3**. Data and error bars represent mean ± SEM, $n = 3$ replicates. Source data are provided as a Source data file.

between E182 of 14-3-3 and the phosphonate group of **3** by engaging in polar interactions with both (Supplementary Fig. 6). The crystal structure compared with the docking pose for **3** revealed a different orientation of its phosphonate, which is rotated around its tetrahedral geometry (by ~109.5°) with the phenylphosphonate group pointing outward of the 14-3-3 central groove (crystal) versus into it (docking; Supplementary Fig. 7). This directs the orientation of the rest of the molecule in the crystal structure, resulting in optimal nestling in the 14-3-3/ChREBP interface pocket (Fig. 3d, e), with the second phenyl beneficially engaging the hydrophobic roof of the groove.

**Small library of analogs establishes crucial SAR.** To study the stabilatory mechanism in more detail, a library around **3** was synthetized and analyzed for SAR. Linker length was found to indeed be essential for the stabilizing activity of **3**, as demonstrated by the inactive derivatives with shorter linkers (**11, 12, 14–16**) and lower EC50 values for slightly longer linker variants (15 or 72 μM for **13** or **17**, respectively) (Table 2, Supplementary Figs. 8–10). A co-crystal structure was obtained for 14-3-3 bound by **12**, one of the inactive short-linker analogs of **3** (Fig. 4a), revealing an identical binding pose to the previously described phosphonate-based inhibitors. Remarkably, with an intermediate linker length ($n = 1$ for $(CH_2)^n$; as noted in Table 1) for **12** compared with **3** ($n = 2$) and the reported inhibitors ($n = 0$), it not only appears to pinpoint the key-determining feature for the mode of action, but additionally hits the 'sweet spot' to turn the switch. A crystallographic overlay of **12** (binding to 14-3-3), with **3** (binding to the 14-3-3/ChREBP binary complex) shows two rotations of the

molecules with respect to each other in their orientation in the binding pocket; around the phosphonate and around the central axis of the phenylphosphonate, which drags the side chain around (Fig. 4b), with this turn in binding orientation resembling a molecular switch between the two distinct modes. Further SAR revealed substitutions of the phenylphosphonic moiety were either not tolerated (Me, **18–20**) or did not significantly enhance the activity (F, **21–23**). The second phenyl on the other hand, was hypothesized to provide an interesting opportunity for structure variations, for which substitutions on all positions might result in engaging the 14-3-3 side chains D215, K122, or N175 (Supplementary Fig. 9). Most substitutions analyzed resulted in similar or slightly improved stabilization (**24–33**). However, the hydrophobic environment engaged by the phenyl does not tolerate a hydroxyl *p*-substitution (**34** is inactive). Placing the hydroxyl group on the linker, however, was allowed, resulting in identical EC50 values of 11.4 μM for both enantiomers (**36** and **37**), which can be explained by their most probable orientation toward the solvent-exposed side as can be observed from the crystal structure. Interestingly, whereas the methylated amide derivative (**38**) is inactive, removing the amide nitrogen (**39**) does not result in the same deleterious effect, suggesting its intramolecular hydrogen bond with a phosphonate oxygen is not essential for the molecule's conformation or stabilizing activity. Two derivatives, a *p*-F-substitution (**26**) and an *o*-OCH2Ph substitution (**30**) showed slightly improved stabilization activities, resulting in a cooperative enhancement of the 14-3-3/ChREBP binding affinity of 26- and 22-fold, respectively (Fig. 4c). Considering that characterization of complex stabilization in solution is dependent on the relative concentrations of the binding partners[39], we collected 2D

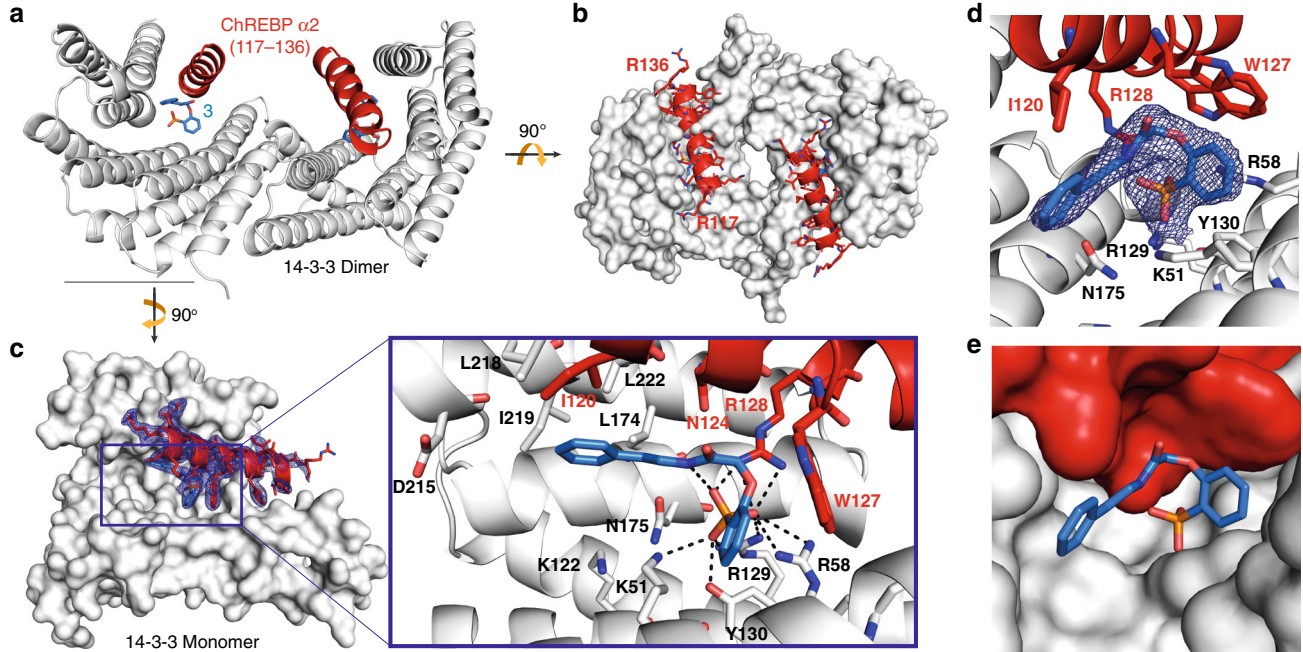

**Fig. 3 Co-crystal structure of 14-3-3β in complex with the ChREBP α2 peptide and stabilizer 3. a** Side view of the 14-3-3 dimer (white cartoon representation) bound by two ChREBP peptides (red cartoon) and two **3** molecules (blue sticks). **b** Top view displaying the antiparallel orientation of the ChREBP α helices in the 14-3-3 dimer. **c** Front view of one 14-3-3 monomer (white surface) bound by one ChREBP α-helix and **3**. Final 2F$_o$–F$_c$ electron density (blue mesh, contoured at 1.0σ) displayed for the ChREBP peptide and **3**. The panel highlights the interactions **3** makes with 14-3-3 and ChREBP residues (relevant side chains are displayed in stick representation). Polar contacts are displayed as black dashed lines. **d, e** Detailed view of the composite pocket occupied by **3** created by ChREBP and 14-3-3 together, illustrating its interface-binding character. Both phenyl rings of **3** make hydrophobic contacts with side chains of both proteins, as observed for the phenylphosphonate with W127 of ChREBP, and the second phenyl with I120 of ChREBP and I219, L222, and L174 of 14-3-3. Residues of both protein partners that make up the binding pocket are represented as cartoon and sticks (**d**) or surface (**e**).

**Table 1 Data collection and refinement statistics (molecular replacement).**

|  | 6YGJ | 6YE9 |
|---|---|---|
| *Data collection* | | |
| Space group | C 1 2 1 | C 2 2 2 1 |
| Cell dimensions | | |
| *a, b, c* (Å) | 98.79, 76.69, 90.29 | 82.84, 112.80, 62.71 |
| *α, β, γ* (°) | 90.00, 119.22, 90.00 | 90.00, 90.00, 90.00 |
| Resolution (Å) | 57.30–2.07 | 34.24–1.80 |
|  | (2.07–2.07)[a] | (1.83–1.80) |
| $R_{sym}$ or $R_{merge}$ | 0.045 (1.198) | 0.116 (0.493) |
| $I / \sigma I$ | 14.8 (1.10) | 9.7 (2.4) |
| Completeness (%) | 99.9 (98.0) | 99.6 (96.7) |
| Redundancy | 6.7 (6.8) | 6.1 (4.6) |
| *Refinement* | | |
| Resolution (Å) | 57.3–2.07 | 34.24–1.80 |
|  | (2.13–2.07) | (1.86–1.80) |
| No. reflections | 35958 (1778) | 27465 (2615) |
| $R_{work}/R_{free}$ | 22.1/27.7 | 17.49/21.92 |
| No. atoms | | |
| Protein | 4108 | 4078 |
| Ligand/ion | 75 | 25 |
| Water | 33 | 309 |
| *B-factors* | | |
| Protein | 74.98 | 16.25 |
| Ligand/ion | 82.64 | 38.79 |
| Water | 58.66 | 28.23 |
| R.m.s. deviations | | |
| Bond lengths (Å) | 0.020 | 0.010 |
| Bond angles (°) | 1.82 | 0.93 |

Datasets of a single crystal for each structure.
[a]Values in parentheses are for highest-resolution shell.

## Table 2 Structure and activity for analogs of 3.

**Structure 1** (m, n variables)

| | | EC$_{50}$ (µM) |
|---|---|---|
| **m = 1** | | |
| 11 | n = 0 | Inactive |
| 12 | n = 1 | Inactive |
| 3 | n = 2 | 9.3 ± 0.7 |
| 13 | n = 3 | 15.1 ± 3.3 |
| **m = 2** | | |
| 14 | n = 0 | Inactive |
| 15 | n = 1 | Inactive |
| 16 | n = 2 | Inactive |
| 17 | n = 3 | 72.5 ± 24.4 |

**Structure 2** (R at positions a, b, c)

| | | pos | EC$_{50}$ (µM) |
|---|---|---|---|
| **R = Me** | | | |
| 18 | | (a) | ~ 30.2 ± 12.2[a] |
| 19 | | (b) | Inactive |
| 20 | | (c) | Inactive |
| **R = F** | | | |
| 21 | | (a) | 16.1 ± 2.2 |
| 22 | | (b) | 9.3 ± 1.5 |
| 23 | | (c) | 12.9 ± 3.0 |

**Structure 3** (X on phenyl)

| | | pos | EC$_{50}$ (µM) |
|---|---|---|---|
| **X = F** | | | |
| 24 | | o | 5.0 ± 0.8 |
| 25 | | m | 10.8 ± 1.4 |
| 26 | | p | 6.4 ± 0.4 |
| **X = CF$_3$** | | | |
| 27 | | o | 5.2 ± 1.3 |
| 28 | | m | 7.3 ± 1.0 |
| 29 | | p | 8.4 ± 0.9 |
| **X = OCH$_2$Ph** | | | |
| 30 | | o | 7.0 ± 0.4 |
| 31 | | m | 3.7 ± 0.5 |
| 32 | | p | 2.3 ± 0.4 |

**Structure 4** (X, Y on phenyl)

| | | EC$_{50}$ (µM) |
|---|---|---|
| **Y = H** | | |
| 33 | X = Cl | 5.7 ± 1.1 |
| 34 | X = OH | Inactive |
| 35 | X = Br | Aggregated |
| **X = H, Y = OH** | | |
| 36 | (S) | 11.4 ± 4.4 |
| 37 | (R) | 11.4 ± 6.1 |

**Structure 5** (N–Z)

| | | EC$_{50}$ (µM) |
|---|---|---|
| 38 | Z = NMe | Inactive |
| 39 | Z = CH$_2$ | 14.9 ± 2.1 |

All compounds were titrated on 2 µM 14-3-3β and 100 nM fluorescently-labeled ChREBP peptide and EC$_{50}$ values were calculated from the resulting dose-response fluorescence anisotropy curves (Supplementary Fig. 10). Data represent mean ± SEM, $n = 3$ replicates. Source data are provided as a Source data file.
[a]No reliable fit.

titration data to investigate the extent of this effect (Supplementary Fig. 11). Titration of compounds **3**, **26**, and **30** to constant peptide (100 nM) and varying protein concentrations revealed a mild effect on EC$_{50}$ values observed (ranges between 6 and 15 µM; 3 and 4.4 µM; and 2.5 and 8 µM for **3**, **26**, and **30**, respectively, for a protein concentration range of 1–50 µM), whereas no significant effect was observed for different fluorescent peptide concentrations (range 50–500 nM).

**Selective stabilization of 14-3-3/ChREBP.** The most potent stabilizers **3**, **26**, and **30**, together with the inactive (i.e., no 14-3-3/ChREBP stabilization activity) short-linker analogs **11** and **12** were studied for their PPI modulatory mode of action and selectivity by titrations on 14-3-3β and five representative client-derived peptide motifs (Fig. 4d). These motifs were selected based on their distinct 14-3-3-binding sequences and included mode I/II (TAZ), mode III (ERα), special mode (p53), and the only other reported non-phosphorylated motif, that of ExoS (Supplementary Table 5 and Supplementary Figs. 12 and 13). First, neither **11** nor **12** was found to influence the binding of p53 and the non-phosphorylated motifs of ChREBP and ExoS to 14-3-3, yet **11** showed PPI inhibition activity for 14-3-3/ERα and 14-3-3/TAZ, which was weaker for **12**. The phenyl in **11** was directly coupled to the amide. Extending this distance with either a methyl (**12**) or ethyl (**3**) linker switched the activity from a weak 14-3-3 PPI inhibitor to a stronger and selective stabilizer for the 14-3-3/ChREBP complex. This 'switch-on' effect is illustrated by the ChREBP-specific EC$_{50}$ of 9.3 µM observed for **3** and the simultaneous abrogation of inhibitory activity against any of the tested 14-3-3 PPIs (Fig. 4d). **26** and **30** were found to have EC$_{50}$ values in the same low µM range for enhancing the binding of ChREBP to 14-3-3 (6.4 and 7.0 µM, respectively), with only very weak inhibitory activity toward mainly TAZ and ERα, with merely hints of inhibition in the high µM range. The inhibitory power toward this set of representative 14-3-3 client motifs is thus absent or much weaker than their stabilizing activity. This demonstrates the highly selective nature of the activity of these compounds by addressing a unique pocket only present in the 14-3-3/ChREBP complex. Together, this data indeed confirms the molecular switch molecular mechanism as observed from the crystal structure (Fig. 4b), and this compound series further shows the evolution and ultimate uncoupling of promiscuous PPI inhibitory toward selective stabilatory activity, based on a highly similar scaffold. This can further be explained by the underlying mechanism of PPI inhibition, driven by the intrinsic affinity of the ligand for 14-3-3 alone to compete with complex formation, compared with cooperative enhancement by binding to a complementary, specific interaction surface, constituted by two protein partners that are engaged simultaneously by a stabilizer. Here, constitution of a ternary complex results in strong stabilization if the small molecule has a low inherent affinity for one (or ideally both) protein partner(s) (low Kd) and high cooperativity (high alpha factor) as mathematically described in previous work[39,40].

## Discussion

We successfully employed an in silico structure-guided strategy to identify selective 14-3-3/ChREBP PPI stabilizers with a phosphonate-based chemotype. The strategy was steered by our previous success in virtual screening for 14-3-3 binders[31]. Structural analysis and SAR revealed the mode of action for stabilization activity as a cooperative molecular glue, by occupying a complementary PPI interface pocket, simultaneously engaging both protein partners. X-ray crystallography, together with biochemical binding data furthermore showed the evolution

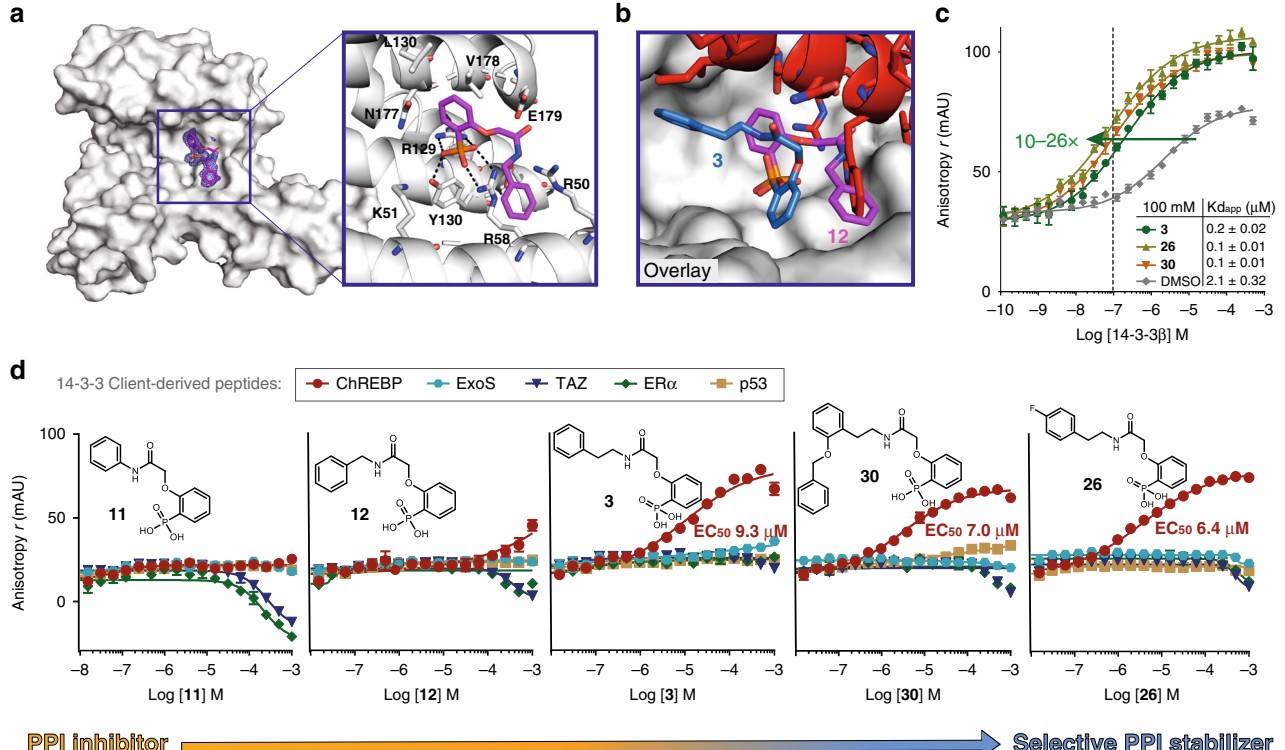

**Fig. 4 Rational optimization of compound series enables selective stabilization of the 14-3-3/ChREBP complex. a** Co-crystal structure of 14-3-3 and **12** binding in the phospho-accepting pocket. Front view of 14-3-3 monomer (white surface) and **12** (purple sticks). Final $2F_o$–$F_c$ electron density (blue mesh, contoured at 1.0σ) displayed for **12**. The panel shows the surrounding amino acid side chains of 14-3-3 (white sticks) and the interactions between **12** and 14-3-3 (black dashed lines). **b** Crystallographic overlay of **12** (purple sticks) binding to 14-3-3, and **3** (blue sticks) binding to the 14-3-3/ChREBP binary complex. **c** Binding curves of 100 nM fluorescently-labeled ChREBP peptide titrated with 14-3-3β in the presence of 100 μM ligands (**3**, **30**, or **26**). Fold stabilization of the protein–protein interaction binding affinity depicted (green arrow). **d** Molecular switch evolution series, starting from PPI inhibitory activity for **11** and **12** (left) toward selective PPI stabilization for **3**, **30**, and **26** (right), as observed from titration data for 14-3-3β (2 μM) and several client-derived peptides (100 nM) each titrated with the compounds. Data and error bars represent mean ± SEM, $n = 3$ replicates. Source data are provided as a Source data file.

of a compound series which revealed a molecular switch mechanism, dissecting weak and promiscuous PPI inhibition from strong stabilization of a specific interaction. Overall, this study demonstrates the principal interchangeability and relatedness of PPI inhibition and stabilization, serving as an inspiration for further efforts toward taking PPI inhibitors and rationally evolving these into stabilizers, potentially even beyond the 14-3-3 realm. Whereas it is especially relevant for hub proteins, such as 14-3-3, this notion in principle holds for many—if not all—globular peptide binding domains (PBDs) like PDZ, SH2, SH3, WW, WH1, PTB that interact with their partner proteins via short, linear (disordered) peptide motifs. Of these, around 1800 are known today[41] that potentially interact with ~100,000 peptide motifs[42]. As such, PPI interfaces—and especially rim-of-the-interface regions—can be targeted by molecules with a potential inhibitor-stabilizer duality that by virtue of small chemical modifications can be directed in either of these two activities. The rational design approach validated here delineates a conceptual entry to the prospective discoveries of small molecules as stabilizers of native protein–protein interactions, which empowers future targeting of hard-to-drug proteins and pathways.

## Methods

**Compound discovery and synthesis.** See Supplementary Methods for detailed descriptions of the virtual screening and molecular docking procedures, organic synthesis and characterization.

**Protein expression and purification.** The full-length (FL) human 14-3-3β protein was expressed from a pPROEX plasmid after transformation to BL21(DE3) competent *E. coli* (Novagen). Cultures were incubated at 37 °C, 140 rpm until $OD_{600} \sim 0.8$ was reached. Protein expression was induced by isopropyl β-D-1-thiogalactopyranoside (IPTG; 0.4 mM) and cells were harvested by centrifugation (10 min, 4 °C, $16,000 \times g$) after overnight expression (18 °C, 140 rpm). Pellets were resuspended in wash buffer (50 mM Tris pH 8.0, 300 mM NaCl, 12.5 mM imidazole and 2 mM β-mercaptoethanol (βME)). After homogenizing the cells (40 bar, Emulsiflex-C3 homogenizer), the soluble fraction was collected by centrifugation (30 min, 4 °C, $40,000 \times g$) and loaded onto a $Ni^{2+}$-affinity column pre-equilibrated with wash buffer. After a washing step (wash buffer + 20 mM imidazole), the bound protein was eluted with 200 mM imidazole followed by dialysis overnight at 4 °C (25 mM HEPES pH 8.0, 200 mM NaCl, 10 mM $MgCl_2$, 0.5 mM tris(2-carboxyethyl)phosphine (TCEP)). The $His_6$-tag of the ΔC variant (14-3-3β truncated after S232) for crystallography was cleaved with TEV-protease during dialysis and subjected to an additional purification by size exclusion chromatography (SEC; Superdex 75; buffer 20 mM HEPES pH 7.5, 100 mM NaCl, 10 mM $MgCl_2$, 2 mM βME). The pure protein was concentrated, aliquoted, flash-frozen in liquid $N_2$, and stored at −80 °C.

Purity and exact mass were determined (Supplementary Fig. 14) using a high-resolution liquid chromatography coupled with mass spectrometry (LC/MS) system comprised of an I-Class Acquity UPLC (Waters) with a Polaris C18A reverse-phase column 2.0 × 100 mm (Agilent), coupled to a Xevo G2 Quadrupole Time of Flight mass spectrometer (Waters). A flow rate of 0.3 mL min⁻¹ was used with a gradient of acetonitrile + 0.1% formic acid (FA) in water + 0.1% FA (acetonitrile 15–75%). Deconvolution of the $m/z$ spectra was done using the $MaxENT_I$ algorithm in the Masslynx v4.1 (SCN862) software.

**Peptide synthesis.** The ChREBP-derived α2 peptide (residues 117–142) was synthesized via Fmoc solid phase peptide synthesis on a TentaGel R Ram resin (Novobiochem; 0.20 mmol/g loading) using an Intavis MultiPep RSi peptide synthesizer. Briefly, Fmoc-protected amino acids (Novabiochem) were dissolved in N-methyl-2-pyrrolidone (NMP, 4.2 eq., 0.5 M) and coupled sequentially to the resin

using N,N-diisopropylethylamine (DIPEA, 8 eq., 1.6 M stock solution in NMP, Biosolve) and O-(1H-6-Chlorobenzotriazole-1-yl)-1,1,3,3-tetramethyluronium hexafluorophosphate (HCTU, 4 eq., 0.4 M stock solution, Novabiochem). Following each consecutive coupling, Fmoc deprotection was performed using 20% piperidine in NMP (1 min, twice). Peptide N-termini were acetylated (Ac$_2$O/pyridine/NMP 1:1:3) or labeled via an Fmoc-O1pen-OH linker (Iris Biotech GmbH) (as previous couplings) with fluorescein-isothiocyanate (FITC; Sigma-Aldrich) (7 eq. with 14 eq. DIPEA), before final deprotection and cleavage off the resin (triisopropylsilane/ethanedithiol (EDT)/water (millipore filtered)/trifluoroacetic acid (TFA), 1:1:1:37, 3.5 h). After precipitation in cold Et$_2$O, peptides were purified on a reverse-phase C18 column (Atlantis T3 prep OBD, 5 μm, 150 × 19 mm, Waters) using a preparative high-performance LC/MS system comprised of an LCQ Deca XP Max ion-trap mass spectrometer equipped with a Surveyor autosampler and Surveyor photodiode detector array (PDA) (Thermo Finnigan). In LC, linear gradients of acetonitrile with 0.1% TFA, in water with 0.1% TFA were used, with a flow rate of 20 mL/min. Fractions with the correct mass were collected using a PrepFC fraction collector (Gilson Inc). Purity and exact mass of all peptides was verified (Supplementary Fig. 15) using analytical LC/MS (C18 Atlantis T3 5 μm, 150 × 1 mm column, 15 min gradient 5–100% acetonitrile with 0.1% TFA in water with 0.1% TFA (LCQ Deca XP Max ion-trap mass spectrometer, Thermo Finnigan).

**Fluorescence anisotropy experiments**. 14-3-3β proteins and FITC-labeled ChREBP α2 peptide were diluted in assay buffer (10 mM HEPES pH 7.4, 150 mM NaCl, 0.1% Tween-20, and 1 mg/mL Bovine Serum Albumin (BSA)). FITC-peptide was used at a final concentration of 100 nM. All compounds were dissolved in dimethylsulfoxide (DMSO, 100 mM stock solutions). Final DMSO in the assay was always 1%. Two-fold dilution series of ligand or 14-3-3β were made in black, round-bottom 384-microwell plates (Corning) in a final sample volume of 10 μL. Fluorescence anisotropy measurements were performed using a Tecan Infinite F500 plate reader (filter set λex: 485 ± 20 nm, λem: 535 ± 25 nm). Reported values are the mean and standard deviation (SD) of triplicates. EC$_{50}$ and apparent Kd values were obtained from fitting the data with a four-parameter logistic model (4PL) in GraphPad Prism 7.

**Isothermal titration calorimetry (ITC) experiments**. 14-3-3β protein and acetylated ChREBP α2 peptide were diluted in buffer (25 mM HEPES pH 7.4, 100 mM NaCl, 10 mM MgCl$_2$, 0.5 mM TCEP, 1% DMSO). The ITC measurements were performed on a Malvern MicroCal iTC$_{200}$. The cell contained 30 μM protein and the syringe 600 μM acetylated peptide. Compound, if present, was at 500 μM. One or two titration series of 18 injections of 2 μL were performed at 25 °C (reference power 5 μCal/s, initial delay 60 s, stirring speed 750 rpm, spacing 180 s). Data for double titrations were merged using ConCat32 software. Data were analyzed in Origin.

**Protein crystallography, X-ray data collection, and refinement**. 14-3-3σ ΔC/**12**. 14-3-3σ ΔC protein was dissolved in crystallization buffer (CB; 20 mM HEPES pH 7.5, 2 mM MgCl$_2$, 2 mM βME) and mixed in a 1:2 molar stoichiometry with compound **12** (100 mM stock in DMSO) to a final protein concentration of 12.5 mg/mL. This was set up for sitting-drop crystallization in a 1:1 ratio in crystallization condition (CB; 0.095 M HEPES pH 7.1, 27% (v/v) PEG 400, 0.19 M CaCl$_2$, 5% (v/v) glycerol) at 4 °C within 4 days. Crystals were fished and flash-frozen in liquid N$_2$. Diffraction data were collected on in-house X-ray diffraction system (equiped with Rigaku MicroMax-003 sealed tube X-ray source and Rigaku Dectris PILATUS3 R 200 K detector). Wavelength of data collection was 1.54187 Å, temperature 100 K. Data were indexed, integrated, and scaled using DIALS[43]. Phases were obtained by molecular replacement using PDB ID 4DHT as search model in Phaser[44]. Coot[45] and phenix.refine[46] were used in alternating cycles of model building and refinement. See Table 1 for data collection and refinement statistics. See Supplementary Fig. 18 for a portion of the electron density map. Ramachandran statistics for this dataset were obtained from the Ramachandran plot: favored/outlier residues 97.86/0.42%, respectively. The structural data for 14-3-3σ ΔC/**12** were submitted to the PDB and obtained entry ID 6YE9.

14-3-3 ΔC/ChREBP-α2/**3**. 14-3-3β ΔC protein, acetylated ChREBP α2 peptide and compound **3** were dissolved in crystallization buffer (CB; 20 mM HEPES pH 7.5, 2 mM MgCl$_2$, 2 mM βME) and mixed in a 1:2:2 molar stoichiometry with a final protein concentration of 12 mg/mL. This was set up for sitting-drop crystallization in a 1:1 ratio with the protein crystallization MPD Suite (Qiagen). Crystals were observed in condition #88 (0.1 M HEPES sodium salt pH 7.5, 30% (w/v) MPD, 5% (w/v) PEG4000) (Supplementary Fig. 16A-C). After initial optimization of pH and PEG4000 concentration, needle-shaped crystals were subjected to the Additive Screen HT (HR2-138, Hampton Research) of which condition #12 resulted in crystals of an improved three-dimensional shape, even though still clustering around a single nucleation site (Supplementary Fig. 16D). The thus obtained crystallization-liquor constitution (0.1 M HEPES pH 7.1, 30% MDP, 1% PEG4000, 0.1 M Ni(II)Cl$_2$•6H$_2$O) was subsequently homemade. Finally, the complex (prepared as described above) was set up for hanging-drop crystallization for crystal reproduction in a 1:1 ratio with the homemade

crystallization-liquor. After 1 day of incubation at room temperature, rod-like clusters were observed (Supplementary Fig. 17A). These were crushed, resulting in small nucleation seeds (Supplementary Fig. 17B) that were subsequently introduced into a fresh drop of pre-equilibrated protein complex in aforementioned crystallization-liquor using a cat whisker (Supplementary Fig. 17C). Single crystals were fished after 1–3 days of incubation at room temperature (Supplementary Fig. 17D), and flash-frozen in liquid N$_2$. Diffraction data were collected at the Deutsches Elektronen-Synchrotron (DESY Hamburg, Germany) PETRA-III beamline P11. Wavelength of data collection was 1.03320 Å, temperature 80 K. The dataset reported was obtained from a crystal that diffracted to a resolution of 2.09 Å. Data were indexed, integrated, and scaled using DIALS[43]. Phases were obtained by molecular replacement using PDB ID 5F74 as search model in Phaser[44]. Coot[45] and phenix.refine[46] were used in alternating cycles of model building and refinement. See Table 1 for data collection and refinement statistics. See Supplementary Fig. 18 for a portion of the electron density map. Ramachandran statistics for this dataset were obtained from the Ramachandran plot: favored/outlier residues 92.96/1.20%, respectively. The structural data for 14-3-3β ΔC/ChREBP−α2/**3** were submitted to the PDB and obtained entry ID 6YGJ.

**Reporting summary**. Further information on research design is available in the Nature Research Reporting Summary linked to this article.

## Data availability

Crystallographic datasets have been deposited in the Protein Data Bank (PDB) and obtained accession codes 6YGJ and 6YE9. Raw data are available from the corresponding authors upon reasonable request. The source data underlying Figs. 1d, 2b and c, 4c and d, and Supplementary Figures 4, 10, 11, 12a, and 13 are provided as a Source data file. Source data are provided with this paper.

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

## Acknowledgements

This work was supported by the Netherlands Organization for Scientific Research (through Gravity program 024.001.035, VICI grant 016.150.366, and ECHO grant 711.018.003) and by DFG-funded CRC1093 (Supramolecular Chemistry on Proteins).

## Author contributions

E.V., C.O., and L.-G.M. initiated the project. P.T. and E.V. designed and performed the virtual screen and docking studies. L.-G.M., P.T., and E.V. selected candidate compounds for in vitro validation. E.S. and E.V. designed and performed binding assays. E.V. purified proteins, synthesized peptides, grew crystals, and solved the co-crystal structure. E.S., C.O., L.B., M.K., and K.P. designed compounds for SAR studies. K.P. performed organic synthesis and characterization of compounds. E.S. designed and performed client-selectivity analysis studies. E.S., E.V., K.P., C.O., M.K., and L.B. interpreted and discussed results. E.S., E.V., C.O., and L.B. wrote the paper with the input from all co-authors.

## Competing interests

L.B. and C.O. are founders and shareholders of AmbAgon Therapeutics. The remaining authors declare no competing interests.
