## [Peer Review File · Nature Communications]

Reviewers' Comments:

Reviewer #1:

Remarks to the Author:

The submission by Sijbesma et al represents a state-of-the-art, interesting study that illustrates that the educated and focused chemical approach allows to gradually transform the PPI inhibitory molecule into a powerful stabilizer. Starting from the recently published structure of the 14-3-3beta complex with ChREBP fragment stabilized by a natural effector AMP, the authors screened for potential molecules that would bind the same pocket as that occupied by AMP and would positively affect the stability of the 14-3-3/ChREBP complex, as analysed by high-efficiency in vitro binding assay (FP). This allowed the authors to pick two hit molecules that caused a modest stabilization, and further evolve one of these hits into a derivative with the 14-fold stabilization of the target interaction (compound 3). The ternary complex of the 14-3-3/ChREBP/3 composition was validated by crystallography at 2.07 Å resolution, permitting detailed analysis of the PPI interface that confirmed a molecular glue mode (contacts with both interacting protein entities). Among the rest, the experimentally determined conformation of cpnd 3 was appreciably different from that predicted by one of the most developed docking algorithms (Glide), one more time indicating the limitations of docking protocols in predicting native ligand binding poses.

Using this background, the authors proceeded with a more detailed SAR analysis and tested overall up to 65 chemical compounds whose synthesis and validation are described in SI (the identity of compounds was confirmed by LC-MS, ¹H-NMR and ¹³C-NMR). This screening allowed to demonstrate both, worsening and improvement of the effect of cpnd 3 in targeting the 14-3-3/ChREBP interaction. Importantly, the three improved stabilizers 3, 26, 30 showed that their stabilizing action on the 14-3-3/ChREBP interaction did not disturb several other 14-3-3 PPIs, implying the selectivity of the stabilizing action.

Overall this data laden study leaves good impression and therefore I recommend publication of its modified version taking into consideration the following minor revisions.

My considerations:

I have problems with the use of word "conversion" in the title. It is ambiguous and reads as if the structure of the specific complex would chemically convert the starting molecule into stabilizer in a reaction of some kind. Consider to replace by for example "evolution".

It is claimed in the abstract and in the final introductory part that the authors did a long way starting from promiscuous PPI inhibitors. In fact, the current beginning of Results appears to miss some important phrases about those inhibitors (what are they? What are the expected constants for them?). As the result, it reads as if the authors would have used screening to find a stabilizer right away. I suggest to expand and clarify this part by emphasizing the start from inhibitors. Also, I suggest to mention a relevant earlier study that showed that phosphate, sulphate and some organic phosphates such as glycerophosphates can modulate the interaction of 14-3-3 with a phosphotarget (PMID 23977325).

The picked 14-3-3/ChREBP interaction is a very unique 14-3-3 PPI in the fact that the 14-3-3 interacting part of ChREBP lacks the typical phosphorylation, which needs to be compensated by inorganic phosphate or some organic phosphates such as AMP. In this respect, although generally positive and encouraging, the approach described by the authors will unlikely guide the creation of other stabilizers from inhibitors for the rest of 14-3-3 PPIs, where the phosphate containing moieties would inevitably compete with target binding. Likewise, the same strategy will unlikely help in the case of ExoS binding, which is phosphorylation-independent but, when bound, structurally clashes with the small molecules used in the study. So, it is most likely a very peculiar case that the authors describe (but still interesting).

I wonder why cpnds 11 and 12, the latter of which was co-crystallized in the present study in the binding pocket, in a conformation not compatible with extra phosphopeptide binding, did not cause any inhibition of the peptides used in Fig. 4d. What is the direct affinity of 12 to 14-3-3?

Can cpnd 3 bind to 14-3-3 directly in vitro? Have the authors tried its co-crystallization with 14-3-3? It seems that differing by only one methylene group, the cpnd 3 and 12 would bind to 14-3-3 in the absence of ChREBP. Then, analysis of their direct binding to 14-3-3 would be informative (I may have missed it!).

Fig. S5 left and right – what is the difference between them? Some info seems missing in the legend.

It is unclear which compounds were synthesized in the study and which were bought (p.14 of SI states that all were purchased but the final pages of SI describes the detailed synthesis).

ABSTRACT, penultimate line – “prospective” seems to be “perspective” or “potential”

RESULTS line 3. A phospho-binding (a typo)

A mixture of British and American English spelling is used: optimization and stabilization, but catalogue, analogue, neighbour. Please use consistently.

Some articles are missing, better to double check with the native speaker.

Nikolai Sluchanko

Reviewer #2:

Remarks to the Author:

This manuscript describes development of a small molecule stabilizer of a protein-protein interaction (PPI). Small molecules that inhibit or stabilize PPIs are difficult to develop – with stabilizers representing the more difficult challenge. The authors have previously described some of the best characterized PPI stabilizers. Here they show that a weak inhibitor of 14-3-3 protein-protein interactions can be judiciously transformed into a stabilizer. Overall, the study is well conceived and the results will be of interest to the chemical biology community.

Some aspects of the manuscript that can be further improved to clarify the message:

1. It is unclear why compound 2 was considered to be more attractive as a starting point for the study. Compound 1 binds with a higher affinity and its covalent reactivity could potentially be tuned.
2. The main concern with the manuscript is the description of the precursor as an inhibitor and the discussion of the “molecular switch” mechanism. In the abstract, the authors state that they succeeded in developing compounds that range “from weak and unspecific inhibition of 14-3-3 PPIs to specific, potent stabilization of the 14-3-3/ChREBP complex.” The challenge with this statement is that the inhibitors (compounds 11 and 12 in Figure 4) don’t inhibit the complex with ChREBP. Indeed, neither 11 or 12 interacts much with 14-3-3/ChREBP. At best, the authors developed stabilizers of 14-3-3/ChREBP from compounds that are extremely weak inhibitors of other complexes of 14-3-3. In this reviewer’s opinion, the manuscript is impactful because the authors were able to rationally design a stabilizer from compounds that mimic phosphates and sulfates in the binding site. The molecular switch terminology distracts from this main message.

Reviewer #3:

Remarks to the Author:

The authors have identified an interesting challenge in molecular design, and they have achieved a level of success that will attract many readers. I believe that this work is ultimately suited to publication in Nat. Comm., but the authors should address a concern mentioned below before the paper is accepted.

For the past several decades, there has been broad interest among academic and industrial researchers in developing synthetic molecules that inhibit protein-protein interactions. As is by now well understood, this type of goal is often challenging to achieve with small molecules because of their size, which is dwarfed by the large surfaces with which many proteins engage one another. Here the authors identify a related but perhaps more complex goal: to develop synthetic molecules that selectively stabilize a particular protein-protein interaction. In general, this type of goal involves similar challenges relative to inhibiting protein-protein interactions. In the case at hand,

the challenge is magnified by the need for selectivity. One of the components, a 14-3-3 protein, has many binding partners, but the authors seek to stabilize the interaction with only one, ChREBP.

The authors employ a variety of techniques to approach their goal, including computational docking, organic synthesis, binding measurements and protein crystallography. The computational screening is driven by a sensible molecular rationale, which begins with inhibitors of interactions involving the 14-3-3 partner. Once a lead is identified, the authors use a logical approach to evolve toward the desired stabilization activity. The function of key compounds is elucidated at atomic level via crystallography.

My principle concern involves the way the authors characterize stabilization of 14-3-3+ChREBP complex formation in solution. The abstract, for example, states that the best compound resulted in "26-fold" enhancement of complex stability. I would imagine that this numerical value is very highly dependent on conditions, specifically, the concentration of each of the protein partners. The authors should provide additional binding assay results to illustrate how the stabilization of the complex varies as the concentration of one partner, or both, is varied (these assays need be performed only with the best small molecule, although additional studies would be valuable). Probably there is a mathematical formalism to explain such variation, and it would be helpful if the authors direct readers to such formalism. However, it is also important for the authors to provide the additional data I suggest, to help readers see what can (and what cannot) be learned from the solution-phase studies employed by the authors to characterize complex stabilization.

Nature Communications manuscript NCOMMS-20-12984

Title: "Structure-based conversion of a promiscuous inhibitor to a selective stabilizer of protein-protein interactions."

Authors: Eline Sijbesma, Emira Visser, Kathrin Plitzko, Philipp Thiel, Lech-Gustav Milroy, Markus Kaiser, Luc Brunsveld, Christian Ottmann

Manuscript revision, June 2020

REVIEWER COMMENTS

Followed by Author response

Reviewer #1 (Remarks to the Author):

The submission by Sijbesma et al represents a state-of-the-art, interesting study that illustrates that the educated and focused chemical approach allows to gradually transform the PPI inhibitory molecule into a powerful stabilizer. Starting from the recently published structure of the 14-3-3 β complex with ChREBP fragment stabilized by a natural effector AMP, the authors screened for potential molecules that would bind the same pocket as that occupied by AMP and would positively affect the stability of the 14-3-3/ChREBP complex, as analysed by high-efficiency in vitro binding assay (FP). This allowed the authors to pick two hit molecules that caused a modest stabilization, and further evolve one of these hits into a derivative with the 14-fold stabilization of the target interaction (compound 3). The ternary complex of the 14-3-3/ChREBP/3 composition was validated by crystallography at 2.07 Å resolution, permitting detailed analysis of the PPI interface that confirmed a molecular glue mode (contacts with both interacting protein entities). Among the rest, the experimentally determined conformation of cpnd 3 was appreciably different from that predicted by one of the most developed docking algorithms (Glide), one more time indicating the limitations of docking protocols in predicting native ligand binding poses.

Using this background, the authors proceeded with a more detailed SAR analysis and tested overall up to 65 chemical compounds whose synthesis and validation are described in SI (the identity of compounds was confirmed by LC-MS, ¹H-NMR and ¹³C-NMR). This screening allowed to demonstrate both, worsening and improvement of the effect of cpnd 3 in targeting the 14-3-3/ChREBP interaction. Importantly, the three improved stabilizers 3, 26, 30 showed that their stabilizing action on the 14-3-3/ChREBP interaction did not disturb several other 14-3-3 PPIs, implying the selectivity of the stabilizing action.

Overall this data laden study leaves good impression and therefore I recommend publication of its modified version taking into consideration the following minor revisions.

We thank the reviewer for the time taken in reviewing our manuscript and for the appreciation of the data presented. The considerations brought forward are insightful and are addressed individually below.

My considerations:

I have problems with the use of word “conversion” in the title. It is ambiguous and reads as if the structure of the specific complex would chemically convert the starting molecule into stabilizer in a reaction of some kind. Consider to replace by for example “evolution”.

Thank you for pointing this out. We agree with your reasoning and have adapted the title of our manuscript now using the word ‘evolution’ instead of ‘conversion’, as you suggested.

It is claimed in the abstract and in the final introductory part that the authors did a long way starting from promiscuous PPI inhibitors. In fact, the current beginning of Results appears to miss some important phrases about those inhibitors (what are they? What are the expected constants for them?). As the result, it reads as if the authors would have used screening to find a stabilizer right away. I suggest to expand and clarify this part by emphasizing the start from inhibitors. Also, I suggest to mention a relevant earlier study that showed that phosphate, sulphate and some organic phosphates such as glycerophosphates can modulate the interaction of 14-3-3 with a phosphotarget (PMID 23977325).

Indeed, the first paragraph of the introduction improved after a re-write and now includes a couple of sentences describing the most important features of phosphate-/phosphonate-based 14-3-3 PPI inhibition and how this served as a starting point for the screening described. We included two additional references of phosphate-based small molecule inhibitors and additionally included a reference to the mentioned study, thank you for your kind suggestion!

The picked 14-3-3/ChREBP interaction is a very unique 14-3-3 PPI in the fact that the 14-3-3 interacting part of ChREBP lacks the typical phosphorylation, which needs to be compensated by inorganic phosphate or some organic phosphates such as AMP. In this respect, although generally positive and encouraging, the approach described by the authors will unlikely guide the creation of other stabilizers from inhibitors for the rest of 14-3-3 PPIs, where the phosphate containing moieties would inevitably compete with target binding. Likewise, the same strategy will unlikely help in the case of ExoS binding, which is phosphorylation-independent but, when bound, structurally clashes with the small molecules used in the study. So, it is most likely a very peculiar case that the authors describe (but still interesting).

We agree with the reviewer that the 14-3-3/ChREBP complex studied here is unique in its constitution, yet we believe this study can serve as an inspiration for further efforts towards taking PPI inhibitors and rationally evolving these into stabilizers, potentially even beyond the 14-3-3 realm. As such, our manuscript is intended to facilitate the consideration of PPI interface – and especially rim-of-the interface – regions as areas in protein complexes that can be targeted by molecules with a (potential) inhibitor-stabilizer duality that by virtue of relative small chemical modifications can be directed in either of these two activities. This consideration is especially relevant for hub proteins like 14-3-3, but in principle for many (if not all) globular peptide binding domains (PBDs) like PDZ, SH2, SH3, WW, WH1, and PTB that interact with their partner proteins via short, linear (disordered) peptide motifs. Of these around 1,800 are known today (Cunningham et al., Nature Methods 2020, 175–183) which potentially can interact with a 100,000 peptide motifs (Tompa et al., MolCell 2014, 55, 161-169). We have reflected this notion now in the concluding section of the manuscript.

I wonder why cpnds 11 and 12, the latter of which was co-crystallized in the present study in the binding pocket, in a conformation not compatible with extra phosphopeptide binding, did not cause

any inhibition of the peptides used in Fig. 4d. What is the direct affinity of 12 to 14-3-3?
Can cpnd 3 bind to 14-3-3 directly in vitro? Have the authors tried its co-crystallization with 14-3-3?
It seems that differing by only one methylene group, the cpnd 3 and 12 would bind to 14-3-3 in the absence of ChREBP. Then, analysis of their direct binding to 14-3-3 would be informative (I may have missed it!).

Indeed, these compounds are extremely weak binders and therefore weak inhibitors, which is exactly what is desired since our aim was to avoid any strong inhibitory effect. Constitution of a ternary complex results in a strong selective stabilization when the small molecule stabilizer has a low inherent affinity for one (or ideally both) protein partner(s) (low K_d) and high cooperativity (high alpha factor) as our group has illustrated previously for 14-3-3 (De Vink et al., ChemSci 2019, 10, 2869-2874) and in this very recent work (Wolter et al, JACS 2020, just accepted manuscript). To help the reader and improve clarity, we have now stronger reflected on this notion in the main manuscript.

Fig. S5 left and right – what is the difference between them? Some info seems missing in the legend.

These two figures are ITC replicates. We thank the reviewer for bringing this forward and we now clarified this in the legend.

It is unclear which compounds were synthesized in the study and which were bought (p.14 of SI states that all were purchased but the final pages of SI describes the detailed synthesis).

Indeed, absolutely right. We have clarified the section in the SI describing which are the bought sets (compounds selected from docking studies including **1** and **2**; and the ‘SAR-by-catalog’ set, compounds **3 – 10**), and which is the custom synthesized set (compounds **11 – 39**).

ABSTRACT, penultimate line – “prospective” seems to be “perspective” or “potential”

RESULTS line 3. A phoshpo-binding (a typo)

A mixture of British and American English spelling is used: optimization and stabilization, but catalogue, analogue, neighbour. Please use consistently.

Some articles are missing, better to double check with the native speaker.

We thank the reviewer for bringing these points to our attention and have carefully revised each of them in addition to paying extra attention to the spelling and word use throughout the manuscript.

Nikolai Sluchanko

Reviewer #2 (Remarks to the Author):

This manuscript describes development of a small molecule stabilizer of a protein-protein interaction (PPI). Small molecules that inhibit or stabilize PPIs are difficult to develop – with stabilizers representing the more difficult challenge. The authors have previously described some of the best characterized PPI stabilizers. Here they show that a weak inhibitor of 14-3-3 protein-protein interactions can be judiciously transformed into a stabilizer. Overall, the study is well conceived and the results will be of interest to the chemical biology community.

We thank the reviewer for the appreciation of our manuscript and its potential impact. The suggestions for further improvement are very insightful and helped us see the main message in a new light. We have aimed to inspire change throughout our manuscript based on the concerns raised, which are addressed in further detail below.

Some aspects of the manuscript that can be further improved to clarify the message:

1. It is unclear why compound **2** was considered to be more attractive as a starting point for the study. Compound **1** binds with a higher affinity and its covalent reactivity could potentially be tuned.

Indeed the chemical liabilities, especially the potential covalent reactivity of **1** and its chemical instability, as well as the synthetically more accessible structure of **2** have guided this decision. The higher initial activity of **1** over **2** was not leading in this decision, in fact, regarding the relative size of these molecules, we were more impressed by the activity of the smaller compound **2**, which has a higher ligand efficiency (LE 0.32 for **2** versus 0.28 for **1**). Covalent modification of 14-3-3 itself might furthermore result in promiscuous inhibition of multiple targets, whereas here we were aiming for selective stabilization for which noncovalent molecules were desirable. Indeed it would also be of potential interest for 14-3-3 modulation to initiate an optimization program starting from **1** yet for the reasons above we didn't further pursue this.

2. The main concern with the manuscript is the description of the precursor as an inhibitor and the discussion of the "molecular switch" mechanism. In the abstract, the authors state that they succeeded in developing compounds that range "from weak and unspecific inhibition of 14-3-3 PPIs to specific, potent stabilization of the 14-3-3/ChREBP complex." The challenge with this statement is that the inhibitors (compounds **11** and **12** in Figure 4) don't inhibit the complex with ChREBP. Indeed, neither **11** or **12** interacts much with 14-3-3/ChREBP. At best, the authors developed stabilizers of 14-3-3/ChREBP from compounds that are extremely weak inhibitors of other complexes of 14-3-3.

We agree with the reviewer this description is not fully correct and furthermore distracts from the main message of our manuscript. We thank you for pointing this out and have rewritten several parts throughout the manuscript to improve on this. In the first paragraph of the introduction, emphasis is placed on the basis of this work, being the phosphate-/phosphonate-based general inhibition of 14-3-3 complexes, which we take as starting point for the development of rim-of-the-interface stabilizers for 14-3-3/ChREBP. Furthermore we have lessened the focus on the molecular switch terminology. As also discussed in response to reviewer # 1, the compounds described in this work are indeed very weak binders to 14-3-3 – enabling strong stabilization upon ternary complex formation due to high cooperativity (high alpha factor), low inherent affinity (low K_d), and as such extremely weakly inhibit 14-3-3 PPIs yet selectively stabilize 14-3-3/ChREBP. We thus agree with the statement in the last sentence of the reviewer of this point 2. To clarify this in the abstract we reworded 'unspecific' into 'general'. Whereas one could argue these compounds are indeed *extremely* weak inhibitors, we respectfully prefer to avoid hyperbolic word use and leave it at 'weak'.

In this reviewer's opinion, the manuscript is impactful because the authors were able to rationally design a stabilizer from compounds that mimic phosphates and sulfates in the binding site. The molecular switch terminology distracts from this main message.

We agree with the reviewer and have adapted the description of the terminology, limiting the 'molecular switch' wording to describing solely the 'turn in binding orientation' of the relative compounds **3** and **12**, while further improving on clarifying the main message throughout the manuscript; being the evolution of a strong stabilizer from promiscuous phosphate-based weak binding/inhibitory molecules.

Reviewer #3 (Remarks to the Author):

The authors have identified an interesting challenge in molecular design, and they have achieved a level of success that will attract many readers. I believe that this work is ultimately suited to publication in Nat. Comm., but the authors should address a concern mentioned below before the paper is accepted.

For the past several decades, there has been broad interest among academic and industrial researchers in developing synthetic molecules that inhibit protein-protein interactions. As is by now well understood, this type of goal is often challenging to achieve with small molecules because of their size, which is dwarfed by the large surfaces with which many proteins engage one another. Here the authors identify a related but perhaps more complex goal: to develop synthetic molecules that selectively stabilize a particular protein-protein interaction. In general, this type of goal involves similar challenges relative to inhibiting protein-protein interactions. In the case at hand, the challenge is magnified by the need for selectivity. One of the components, a 14-3-3 protein, has many binding partners, but the authors seek to stabilize the interaction with only one, ChREBP.

The authors employ a variety of techniques to approach their goal, including computational docking, organic synthesis, binding measurements and protein crystallography. The computational screening is driven by a sensible molecular rationale, which begins with inhibitors of interactions involving the 14-3-3 partner. Once a lead is identified, the authors use a logical approach to evolve toward the desired stabilization activity. The function of key compounds is elucidated at atomic level via crystallography.

My principle concern involves the way the authors characterize stabilization of 14-3-3+ChREBP complex formation in solution. The abstract, for example, states that the best compound resulted in "26-fold" enhancement of complex stability. I would imagine that this numerical value is very highly dependent on conditions, specifically, the concentration of each of the protein partners. The authors should provide additional binding assay results to illustrate how the stabilization of the complex varies as the concentration of one partner, or both, is varied (these assays need be performed only with the best small molecule, although additional studies would be valuable). Probably there is a mathematical formalism to explain such variation, and it would be helpful if the authors direct readers to such formalism. However, it is also important for the authors to provide the additional data I suggest, to help readers see what can (and what cannot) be learned from the solution-phase studies employed by the authors to characterize complex stabilization.

We kindly thank the reviewer for the effort invested in carefully revising our manuscript and appreciate their praising words. The concern raised is a fair one and we thank you for pointing out this relevant aspect in the quantification of the degree of PPI stabilization. The fold-enhancement in affinity of two protein partner towards each other, via a small molecule binding to the secondary complex, obtaining an overall more stable ternary complex, is indeed dependent on the concentrations in the assay. We have added the binding data the reviewer fairly requested, for

compounds **3**, **26**, and **30** to the SI (new Figure S11) and have reflected on the results in the manuscript. Regarding the mathematical formalism to describe such variation when characterizing PPI stabilization and in line with a response to reviewer # 1; constitution of a ternary complex results in a strong selective stabilization if the small molecule stabilizer has a low inherent affinity for one (or ideally both) protein partner(s) (low K_d) and high cooperativity (high α factor) as our group has illustrated previously for 14-3-3 (De Vink et al., ChemSci 2019, 10, 2869-2874), and described in further detail in a very recent work (Wolter et al, J Am Chem Soc. 2020 Jun 5. doi: 10.1021/jacs.0c02151.). To help the reader and improve clarity, we have now stronger reflected on this notion in the main manuscript and direct the readers to the formalisms described in the mentioned references.